# Driving Innovation Through Regulatory Design and Corporate Behaviour: A Case Study of Functional Food Industry in Japan

**DOI:** 10.3390/foods13203302

**Published:** 2024-10-18

**Authors:** Keigo Sato, Kota Kodama, Shintaro Sengoku

**Affiliations:** 1Department of Innovation Science, School of Environment and Society, Institute of Science Tokyo, 3-3-6 Shibaura, Minato-ku, Tokyo 108-0023, Japan; sattu0909@gmail.com; 2School of Pharmacy and Pharmaceutical Sciences, Hoshi University, 2-4-41 Ebara, Shinagawa-ku, Tokyo 142-8501, Japan; kodama.kota@hoshi.ac.jp

**Keywords:** functional foods, dietary supplements, foods for specified health uses, foods with function claims, regulation, innovation

## Abstract

This study addresses the critical need for innovation in the healthcare sector, particularly in Japan’s functional food industry, amid rising chronic diseases and healthcare costs. It explores the complex relationship between regulatory design and corporate behaviour, focusing on how companies’ compliance strategies influence their research and development (R&D) investments. A mixed-method approach was used, analysing data from 15 major dietary supplement companies and 74 products under Japan’s Foods with Function Claims (FFCs) regulation. The study reveals a correlation between companies’ engagement in the FFCs system and their preference for conducting in-house clinical trials, indicative of higher R&D investments (*R* = 0.66, *p* = 0.007), and that between the latter variable and average product sales, which is a measure of returns on regulatory compliance (*R* = 0.66, *p* = 0.008). Companies actively complying with FFCs regulations tend to conduct R&D and accumulate knowledge in-house, to innovate and differentiate their products, gaining competitive advantages. The study also highlights the role of a company’s size, market presence, and industry origins in shaping regulatory strategies, with firms from other industries using FFCs regulations to explore new business opportunities. The research underscores the importance of flexible regulatory frameworks that encourage R&D investment, leading to innovation and competitive advantages in the healthcare sector.

## 1. Introduction

### 1.1. Background

Owing to the increase in chronic diseases and the rising cost of medical care, innovation is expected in the medical and healthcare fields for low-cost, highly effective medical care [1,2,3]. In the medical and healthcare industries, regulations ensure accountability for both performance levels and value for money, as well as improving performance and quality [4]. In reality, the government of each region regulates these products and services to focus on public interest rather than allowing the market mechanism to function autonomously. However, existing regulations related to product safety, efficacy, and quality have focused on protecting consumers and avoiding market failure; in some cases, they have inhibited innovation [5]. However, Christensen [6] argued that an appropriate relationship between regulation and innovation is essential for dramatic improvements in cost and access and concluded that deregulation should not only aim to promote competition but aim to promote competition with disruptive innovation [6]. The effectiveness of a strategy to initiate innovation from places beyond the reach of regulation was further argued based on examples of disruptive innovation occurring in markets that are on the periphery of the regulatory domain and subsequently breaking down regulations in the domain [6]. In this context, there is a growing number of innovations in the relatively weakly regulated healthcare domain, on the periphery of the strongly regulated medical domain, aimed at extending life expectancy, improving quality of life, diagnostic and treatment options, and increasing the efficiency and cost-effectiveness of the healthcare system [7]. Recent regulatory reforms implemented by the U.S. Food and Drug Administration have spurred the growth of mobile health (mHealth) products by setting leading guidelines and reforming regulatory systems and structures [8,9]. This finding suggests that properly designed regulations can promote product and process innovation.

Unlike the medical sector, the outer healthcare sector assumes that products and services are traded in a competitive, free-market environment. Regulation in the healthcare industry has focused on preventing market failures, especially those caused by information asymmetries, and indirect regulation has been established. However, compared with the medical domain, the newly formed healthcare domain has not yet fully developed appropriate regulations [8,10]. This is because, at the dawn of a newly emerging industry, it started with no institutions and regulations, followed by the gradual establishment of appropriate regulations [11]. Thus, there is a risk of market failure, including disadvantages for consumers and impediments to market formation.

The appropriate institutional design of regulations is important in the healthcare sector to achieve consumer protection, while allowing companies to innovate in free competition and promote the healthy development of the industry. In mHealth and other fields, regulatory design coevolves with the technology [8,12]. Inducing innovation by setting regulations to encourage companies to function effectively is also a point of contention from a science innovation perspective [13,14].

### 1.2. Research Objectives

This study aims to clarify the effects of regulations and corporate behaviour on innovation by analysing company behaviour toward the system from the perspective of clinical trials as a research and development (R&D) investment. Furthermore, recommendations should be made on how a system should be designed to create a sound healthcare market. Functional food includes dietary supplements, foods in normal form, and beverages. Dietary supplements have the characteristics of both pharmaceutical and consumer products; the pharmaceutical regulations influence their regulations, and their global market size is large. This industry is appropriate for discussing the relationship between regulation and innovation, as detailed in the next section. With this objective, the dietary supplement industry in Japan was the focus of this study.

## 2. Literature Review

### 2.1. Impacts of Regulation on Innovation

Regulations are broadly classified as economic, social, or institutional [15,16,17]. Historically, regulations aimed to counter market failures and externalities, potentially constraining firm options and profits [18,19]. Although regulations are designed to meet certain goals, they exhibit a multifaceted impact on corporate behaviour and innovation. The impact of regulation on innovation has both restrictive and promotive aspects. Regulatory compliance escalates costs for firms, limits R&D investment resources, and hinders technological advancement [19]. Regulations, particularly those mandating performance and safety standards, may prevent firms from creating new products [20,21]. Market entry regulations are negative for industry-wide innovation, because they discourage market entry by innovative new entrants [16]. In this case, deregulation promotes innovation by lowering the barriers to entry. Conversely, stricter regulations can drive innovation. For instance, stringent regulations in the environmental sector have prompted technological development and product launches [14,22]. The “Porter hypothesis” [14] suggests that appropriately designed environmental regulations stimulate technological innovation, leading to cost efficiencies and quality enhancements that offset compliance costs [23]. The Porter hypothesis was controversially criticized by neoclassical economists, such as Palmer, who argued that if it is possible to increase profits by tightening environmental regulations, no rational firm would structurally miss such an opportunity [13]. The Porter hypothesis has since been tested in various cases. Some studies have demonstrated that environmental regulations can have a positive long-term influence on innovation, even when they negatively impact short-term results [16,17,18,19].

Factors influencing the relationship between regulation and innovation have been examined, with a particular focus on environmental regulations because of their societal significance [24,25]. Outside environmental regulations and the relationship between social regulations and innovation have only been analysed in a limited number of areas. While product safety regulations can constrain innovation by imposing compliance burdens, they also incentivize inventive product development to gain consumer favour [16,17]. Regulations that encourage flexible approaches, such as incentive-based regulations and performance standards, tend to boost innovation [14,17]. Complete market information promoted by regulations reduces information asymmetry and fosters innovation [16,26].

The effects of regulations on innovation vary according to industry characteristics, firms, and technologies, thus necessitating well-designed regulations to foster innovation [27]. Numerous regulatory and firm factors influence the intricate relationship between regulation, innovation, and firm behaviour. While compliance costs may erode the resources available for R&D and discourage innovation, they may also induce more efficient R&D. Firms would gain a competitive advantage by complying with regulations, extending their knowledge, adopting technologies, and developing technologies, processes, and products. Incentives for compliance with strict regulations may arise from competitive advantages through direct compliance and indirect benefits, such as enhanced productivity. Firms navigate these complexities by balancing compliance costs with incentives for technological development.

However, a comprehensive understanding of this relationship remains elusive because of the diverse factors affecting it. While many studies have adopted industry-level data to investigate this link, few have examined individual firm behaviour and innovation.

Moreover, despite the importance of social regulations and high expectations for innovation in the healthcare sector, there is a lack of research on the relationship between regulations and innovation. Case studies on the relationship between regulations, firm behaviour, and innovation in this area could provide useful material for theory building.

### 2.2. Functional Food

This study delineates the exploration of functional foods, with a particular emphasis on dietary supplements. Among orally ingested products, pharmaceuticals represent the medical domain; whereas, food represents the nonmedical domain. Unlike traditional foods, functional foods contain functional physiological components that are anticipated to affect the human body. They hold promise for enhancing health, aiding the prevention of diseases in ageing societies, potentially reducing the strain on public health infrastructure, and are under consideration for self-medication utilization [28,29,30]. For this manuscript, the term “functional foods” denotes foods that offer health benefits that exceed their nutritional value.

Functional foods are subject to specific legal frameworks in various nations [31]. Regulations vary from country to country, and international harmonization has not been achieved. Generally, these regulatory frameworks encompass product categorization, morphological descriptions, intended use, approved lists, good manufacturing practice (GMP) systems, and intellectual property considerations. Analogous to pharmaceuticals, these regulations strive for quality, safety, efficacy assurance, information disclosure, and appropriate labelling [32,33,34,35,36,37,38,39]. These regulations aim to safeguard consumers and maintain competitive market integrity [40,41,42,43].

The nuanced effects of functional ingredients in dietary supplements pose challenges to consumers in discerning the potential health advantages of functional food consumption. Despite consumers expressing keen interest in benefit-related information [44,45,46], they often lack a comprehensive understanding, leading to information asymmetry. Unreliable information can hinder consumer decision making. Consequently, information regarding these health benefits is stringently regulated in terms of presentation and labelling.

Securing health claims imposes various costs on companies, including research and development expenses to validate the long-term health advantages of functional foods, along with administrative processing costs [47]. Conversely, obtaining health claims can augment a product’s value [44,45]. In the functional food sector, newcomers may produce analogous functional foods cost-effectively, given the challenges in patenting food ingredients, in contrast to pharmaceutical components that enjoy patent protection [48]. These dynamics can prevent established manufacturers from investing in novel food functionality research. Hence, Hobbs et al. suggested that health claims labelling restrictions could potentially inhibit free riders, thereby fostering innovation [49]. However, rigorous regulations might deter development efficiency, inflate product development and health claim costs, and stimulate innovation and market competitiveness [47]. Administrative expenses can vary depending on the process of health claim acquisition, with categories including notification and approval [31].

### 2.3. Functional Food Market and Regulation in Japan

This study selected the Japanese functional food sector, which is located between the medical and nonmedical sectors. Japan’s functional food market, estimated at USD 13 billion, coupled with its dietary supplement market valued at USD 6 billion, positions itself among global leaders [10,50]. Recent regulatory changes have created dynamics in market and business behaviour and have attracted attention from a global perspective. The “Foods with Health Claims (FHCs)” system serves as the regulatory framework for functional foods. In the market landscape, products that adhere to the FHCs system coexist with their noncompliant counterparts. The dietary supplement industry straddles the boundaries of the FHCs regulations and is characterized by existing at a larger scale outside the regulation. Additionally, the FHCs system was segmented into Foods for Specified Health Uses (FOSHUs), Foods with Nutrient Function Claims (FNFCs), and Foods with Function Claims (FFCs). The FNFCs segment operates as a self-certification mechanism that primarily focuses on conventional nutrients. In contrast, the FOSHUs and FFCs paradigms pertain to the health claim labelling of food ingredients, surpassing these conventional nutrients. Table 1 presents an overview of Japan’s regulatory landscape for dietary supplements and pharmaceuticals.

The FFCs framework, instituted in 2015, was envisioned to invigorate the health food market through FOSHUs deregulation, thereby enabling the introduction of food products with potential health benefits, alleviating lifestyle-related ailments, and reducing consumer healthcare expenses [51]. Although such functional food labelling relies on scientific validation, the obligation for accuracy resides with food business entities [29,52,53].

From an industrial perspective, deregulation aims to minimize corporate development costs and risks [51,54]. Under the FFCs paradigm, the Japanese government has abstained from assessing safety and efficacy, thus streamlining the process by endorsing a notification system in which operational responsibility is an integral component of administrative procedures. This deregulation allowed many companies, mainly small retailers, to enter the Japanese FHCs market and contributed to an increase in the diverse range of companies entering the market, thereby broadening its base. This scheme also contributes to the growth of existing companies [10].

To evaluate product functionality, companies can adopt two methodologies: a systematic review (SR) of academic publications detailing the clinical trials (CTs) of a product’s ingredients or a direct CT of the product in question [10,53]. The SR approach, which leverages external knowledge, is a cost-effective alternative to manufacturers’ in-house CTs. Small companies also tend to use SR [10]. However, the introduction of multiple similar FFCs products based on identical SRs into the marketplace by various companies may erode the competitive edge of those following the SR evaluation route. In contrast, products assessed via the CT pathway tend to be more resource-intensive than their SR counterparts, primarily because CT-based products are executed in-house.

In previous studies, top companies registered robust sales by combining functional materials, conducting their own CTs, nurturing internal knowledge repositories, and differentiating their products by creating superior functional products [10]. To increase their market product stature, these entities would intensify requisite R&D undertakings, including multimaterial formula development and proprietary testing protocols.

When comparing the regulations for dietary supplements and pharmaceuticals, we observed similarities in the standards of quality, safety, and efficacy, as shown in Table 1. Dietary supplements incorporate regulatory elements traditionally associated with pharmaceuticals, including GMPs and CTs.

## 3. Research Question

The restrictive and promotive impacts of regulation on innovation have not been discussed much in the context of the healthcare sector, as discussed in the previous section. This study focuses on the influence of regulation on the innovation of functional foods. We delved deeper into the implications of these regulations, focusing on how they influence corporate behaviour and innovation. In particular, we examined existing companies that produce FFCs products and their strategic decisions, especially regarding in-house CTs and associated R&D investments.

Regulatory conformity to the FFCs would provide companies with a competitive advantage, such as increased product value and quality, by labelling health claims. Regulatory compliance costs are lower under the SR route, which utilises external knowledge without the cost and time of CTs. In this case, regulatory compliance would occur even with a smaller return from compliance because of lower compliance costs. Conversely, if the return on compliance is high, a company can invest in R&D with high compliance costs.

While companies must pay costs to comply with the regulations, the benefits gained from compliance provide an incentive for companies to comply with the regulations. It was assumed that companies would decide the appropriateness of regulatory compliance in terms of cost-effectiveness based on the costs of regulatory compliance and regulatory compliance incentives. The balance between regulatory compliance costs and incentives depends on a firm’s attributes and its position within and outside the industry.

Based on the above theoretical development, the following research questions were stated.

Is the firm’s regulatory response behaviour related to its R&D activities?What factors influence the firm’s regulatory response strategy?

To discuss the relationship between regulations and firms’ behaviour to obtain the implications for designing regulations that are compatible with innovation, observing Japan’s multitrack FFCs regulatory environment is a suitable case study. It provides a variety of regulatory strategies for companies. Japan’s market is expanding because of regulatory transitions in functional foods, the entry of small- and medium-sized firms, and the launch of high-sales products by existing firms. Major firms have launched innovative products with novel health claims using CTs. In this study, we focus on the relationship between regulatory responses and R&D for major firms utilizing FFCs regulations and empirically and quantitatively analyse the impact of regulations on firm behaviour using data from companies and products. We posit that the study of the Japanese FFCs system is relevant to the discussion about the achievement of optimal regulation in the healthcare sector, innovation promotion and inhibition in a regulatory environment, and how a properly implemented regulatory healthcare reform can stimulate innovation in technology and the delivery of healthcare. To answer these research questions, we analysed the attributes of dietary supplement companies producing FFCs and their products.

## 4. Materials and Methods

### 4.1. Analysis 1: Statistical Analysis of FFCs Dietary Supplement Companies

For statistical analysis, a dataset of 15 major FFCs dietary supplement companies was created as follows.

The top 30 dietary supplement manufacturers were identified using a comprehensive company-by-company disaggregation of their product sales in 2020 from an industry information data book that surveyed the entire dietary supplements market [55]. The annual sales of these companies’ dietary supplements totalled JPY 590.7 billion per year (approximately USD 4 billion), covering 61% of the Japanese dietary supplement market.

Next, data on FFCs products were explored for the top 30 manufacturers, using (a) the CAA database of FFCs [56] and (b) relevant market research data [55] as sources of information. As a result, a dataset for 74 FFCs products sold by 15 companies was created, as shown in the supplemental data in the previous study [10], excluding one company for which sales data for 2015 could not be obtained and 14 companies without FFCs [55]. These 74 FFCs products consisted of those for which sales [55] could be identified among the FFCs products [56] submitted from 1 April 2015, when the system started, to 31 March 2022.

These 15 companies’ products consist of dietary supplements with functional ingredients, such as ingredients derived from plants and animals, as well as those with vitamins and minerals. It is noted that the Japanese market has a large share of the former dietary supplements, and vitamins and minerals are classified in the FNFCs segment.

Finally, a dataset of the 15 companies selling FFCs products consisting of the following variables was created by sorting these 74 FFCs product datasets and adding the firms’ sales and attribute data.

The variables included the following:Company sales (only data available);Total sales of dietary supplements in 2015 (when the FFCs system started) and 2020;The compound annual growth rate (CAGR) between 2015 and 2020 (calculated using the following formula: CAGR = (Sales in 2020/Sales in 2015)^0.2^ – 1;Total sales of FFCs;Total sales of in-house CTs of FFCs (in 74 FFCs product datasets, FFCs were divided into the following two categories: in-house clinical trial type (in-house CT)—products for which clinical trials are being conducted in-house—and external clinical trial type—products that have not undergone clinical trials in-house);Number of FFCs;Sales composition rate of FFCs (calculated by dividing total sales of FFCs by total sales of dietary supplements);In-house CT rate (calculated by dividing the total sales of in-house CTs of FFCs by the total sales of FFCs);Average sales per FFCs product (calculated by dividing total sales of FFCs by the number of FFCs products).

Statistical analyses were performed using R statistical software (version 3.4.1, R Foundation, 2017).

### 4.2. Analysis 2: Observation of 15 Companies’ Regulatory Response Strategies

To gain further insight into the characteristics of firm behaviour, the regulatory compliance and innovation behaviour of 15 firms were observed. Information on each company was collected from the company’s website.

## 5. Results

### 5.1. Analysis 1: Relationship between Regulatory Response and R&D Activities

Table 2 presents the properties of the FFCs companies (n = 15) and delineates the features of their products. Notably, several manufacturers that focus primarily on dietary supplements are privately owned entities; hence, specific sales data remain undisclosed. For major companies, revenue from the dietary supplement sector constitutes less than 10% of total sales.

Table 3 provides a detailed statistical overview of the characteristics and properties of FFCs companies and their products.

Table 4 reveals significant correlation coefficients among various indicators for the 15 firms that have adopted the FFCs system. A robust correlation was observed between the total sales of dietary supplements, FFCs sales, and in-house FFCs sales, with values ranging from *R* = 0.72 to *R* = 0.95.

Hereafter, three vital variables were noted:

(a) The sales composition rate of FFCs products, which ranged between 0.3% and 100%, suggests that FFCs companies exhibit considerable variability in their engagement with the FFCs system.

(b) The in-house CT rate showed a significant correlation with the FFCs rate (*R* = 0.66, *p* = 0.007).

(c) Average sales per FFCs product correlated with the FFCs rate (*R* = 0.51, *p* = 0.050) and the in-house CT rate (*R* = 0.66, *p* = 0.008).

Multiple regression analysis was performed with the in-house CT rate as the objective variable and explanatory variables, including the sales composition rate of FFCs products and average sales per FFCs product. Because the maximum of all correlation coefficients was *R* = 0.66, showing loose positive relationships, there was little concern about multicollinearity. The results of the multiple regression analysis showed that the coefficient for the sales composition rate of FFCs products was 0.53 (*p* = 0.069), and the coefficient for average sales per FFCs product was 0.062 (*p* = 0.071) (Table 5).

These findings imply that companies with more active engagement in the FFCs system (reflected by a higher FFCs sales ratio) generally prefer to conduct in-house clinical tests to foster the development of high-sales products. Conversely, companies that adopt the FFCs system to a limited extent might be strategizing to capitalize on external expertise, thereby curating an array of FFCs products that are more economically viable, owing to reduced compliance costs.

### 5.2. Analysis 2: Companies’ Regulatory Response Strategies and Factors Affecting Them

Company #1 (Suntory) is a subsidiary of a major food and beverage manufacturing group with sales exceeding JPY 2 trillion [57]. The sales composition rate of FFCs was 35%, which is close to the average for all 15 companies (38%). The three FFCs products included in this study were in-house CT types, and all had acquired new functional claims, such as joint care and brain health. In response to the FFCs regulations, the company complied with the regulations, invested in innovation, and differentiated competitive products. It is presumed that a small number of large products with competitive advantages are being developed. Conversely, the remaining products (such as those that are labelled “anti-ageing”) are marketed as “so-called health foods” and avoid regulation.

Companies #2 (DHC) and #4 (FANCL) have cosmetics and dietary supplements as their business lines. Their channels are both mail-order and retail. The corporate strategies of #2 and #4 are quite different. Company #2 declared that it would provide a variety of products at low prices as a part of its corporate policy [58]. The sales composition rate of the FFCs was 15%, and the in-house CT rate was 0%. Owing to its low-cost strategy, its R&D investment was estimated to be low, and the company was reluctant to respond to FFCs regulations with compliance costs. It was presumed that their strategy was to concentrate on products that could obtain new functions; utilise SR, which is external knowledge; partially comply with regulations with low compliance costs; and efficiently gain a competitive advantage. By contrast, company #4 announced that it would actively utilise FFCs as a corporate strategy [59]. The number of FFCs products covered in this study was 16, which was the largest among the 15 firms. Both the sales composition rate of FFCs and the in-house CT rate were higher than those of FFCs #2. It was presumed that they aimed to actively comply with the FFCs regulations, invest in innovation, and differentiate themselves to obtain competitive advantages.

Company #6 (Meiji) is a major dairy, confectionery, and pharmaceutical manufacturer [60]. The company’s main dietary supplement is a sports-oriented protein, primarily whey protein derived from milk. Under the FFCs system, basic nutrients such as proteins are out of scope, so most products of the company are not FFCs, with a low sales composition rate of FFCs at 6%. Promotions involving athletes include selling their protein products as large brands in retail channels. However, the company conducts its testing of amino-acid-based ingredients and markets them as in-house CT-type FFCs dietary supplements for sports. In addition, it should be noted that in the company’s pharmaceutical business, based on synergies between the group’s pharmaceutical and food businesses, the company also sells dietary supplements to the medical institution channel.

Company #8 (Asahi) is a beverage and food manufacturer, and #10 (Otsuka Pharmaceutical) is a pharmaceutical and nutritional food manufacturer [61,62]. Both companies sell a series of brands of dietary supplements, such as vitamins, minerals, and herbs, primarily through store channels. These “traditional” dietary supplement lineups have remained in the two companies’ FFCs sales at a rate of approximately 10%. Company #8 also developed products based on the seeds of research on lactic acid bacteria and fermented milk ingredients from a lactic acid bacteria beverage manufacturer acquired in 2012. This in-house CT-type FFCs product provides mental health and BP functions, thereby increasing the in-house CT ratio to 59%.

Companies #11 (Egao), #14 (Kyusai), #15 (Setagaya Shizen-shokuhin), and #18 (Yazuya) are dietary supplement manufacturers that specialize in mail-order sales [63,64,65,66]. Whereas #11 and #18 had high FFCs rates (51% and 87%, respectively) and high in-house CT rates (93% and 100%, respectively), #14 and #15 had low FFCs rates (0.3% and 4%, respectively) and low in-house CT rates (0% and 26%, respectively). One common feature of #11 and #18 is that their main products are joint-care dietary supplements, which they market as in-house CT-type FFCs. Joint care is a new health claim that does not exist in the FOSHUs, and granting a health claim can strongly differentiate a product. This may be why FFCs products are offered through in-house CTs with R&D investment.

Company #13 (Ajinomoto) is a seasoning and food manufacturer [67]. Their products have been used in the research and manufacture of amino acids for several years. Dietary supplements consist of non-FFCs products taken in the sports scene and FFCs products for health purposes, with an FFCs rate of 56%. Non-FFCs products have been branded by appealing to the diversity of amino acid formulas and using athletes in their promotions. The latter products are also made of amino acids, with health claims such as “sleep improvement” and “muscle function maintenance”, which were not included in the FOSHUs, based on their research.

Companies #16 (Kobayashi Pharmaceutical) and #20 (Taisho Pharmaceutical) are manufacturers dealing with over-the-counter medicines and daily necessities [68,69]. They have a series of brands with a large lineup of dietary supplement products and FFCs sales composition rates of 4% and 40%, respectively.

Company #22 (Fujifilm) is a manufacturer with business sectors in chemicals, electronics, and medical devices, and #25 (Lion) is a manufacturer with business sectors in toiletries and daily necessities, entering the dietary supplement market in the 2000s under a diversification strategy [70,71]. Both companies sold in-house CT-type FFCs dietary supplements based on their original R&D, resulting in high FFCs rates (100% and 85%, respectively) and high in-house CT rates (98% and 100%, respectively). These main dietary supplements are both FFCs with new functions related to weight loss that the FOSHUs system does not possess.

Table 6 summarizes companies’ behaviour for FFCs regulation, as described above. It can be inferred that the sales composition rate of FFCs and the in-house CT rate would be affected by competitive strategies, product lineup, brand strength, product area in efficacy, and technology seeds.

Figure 1 shows 15 companies’ (a) sales composition rate of FFCs and (b) in-house CT rate, as shown in Table 2. The bold line shows a linear approximation of (a) and (b). The graph was divided into four quadrants based on the respective 50% of (a) and (b), indicated by the dashed lines. The following is an overview of the four quadrants.

The bottom left group of firms, with numerous brand lines (#16, #20) or “traditional” dietary supplements (#10), are less active in R&D and less compliant with FFCs regulations. Some firms (#14 and #15) have mail-order business models.

Firms in the upper left quadrant have low FFCs regulatory compliance but are active in R&D activities. Firms #1 and #6 are both large food companies and have products that are not covered by FFCs regulations (such as those that are labelled for “sports use” or “anti-ageing”) but with clinical evidence.

Meanwhile, the group of companies in the upper right quadrant have created unique products through R&D and further differentiated themselves with novel health claims (“weight loss” or “joint care”) through compliance with regulations. In particular, #22 and #25 are dietary supplement businesses that have entered the market from different industries.

In the lower right quadrant, there were no firms with a lineup centred on FFCs products with SR.

## 6. Discussion

### 6.1. Competitive Strategy with Interlinked Both Regulatory Response and R&D Efforts

As explained in Section 5.1, the relationship between the sales composition rate of FFCs products and the rate of in-house CTs indicates a link between firms that vigorously adhere to FFCs regulations and those that seek to enhance product differentiation and competitive advantages by developing products rooted in their own CTs. Considering (a) sales composition rate of FFCs products as a metric for compliance with FFCs regulations, (b) in-house CT rate as an indicator of R&D activity as additional efforts for compliance, and (c) average sales per FFCs product as a measure of returns on regulatory compliance, our analysis has concluded the following, illustrated in Figure 2.

As shown in the right column in Figure 2, companies aiming to actively utilise FFCs regulations often create knowledge in-house and strive for product differentiation, thereby gaining a competitive advantage. Such distinctiveness allows these companies to record substantial sales, offsetting the elevated compliance expenditures linked to aggressive R&D investments and in-house CTs. Consequently, firms may opt for in-house CTs to innovate within regulatory confines and craft products that are unique, attractive, and offer premium value. The rate of in-house CTs serving as a proxy for R&D endeavours epitomizes a company’s intensified efforts to align with regulations and maximize potential benefits in the regulatory landscape.

Conversely, as shown in the middle column in Figure 2, firms adopting partial or passive approaches to FFCs regulations cultivate product portfolios that garner modest sales by leveraging external knowledge resources. SRs, which draw on collective research efforts over a duration often surpassing firms’ CTs [10], have substantially supported these endeavours, effectively translating amassed knowledge from academic domains into practical products in the market.

In terms of health claims, while compliance costs comprise administrative expenses (e.g., notification) and R&D expenditures (for CTs), the impetus for compliance stems from the augmentation of product value through health claims and the accrual of competitive advantages via product differentiation. It can be posited that companies gauge the judiciousness of regulatory compliance based on cost-effectiveness and weigh compliance costs against incentives to comply. This concept of “efficiency related to regulatory compliance” can be perceived as an underlying construct influencing regulatory adherence.

However, as shown in Table 4, no significant relationship was observed between in-house clinical trial rates and metrics, such as revenue and CAGR (with *R*-values of 0.13 and 0.29). Such outcomes indicate that, when comparing strategies that rely on external resources versus in-house development, neither approach exhibits a pronounced advantage in terms of company performance.

### 6.2. Impact of Companies’ Characteristics on Strategies of Regulatory Response and R&D

This section provides an overview and integrated discussion of R&D and regulatory response strategies based on Figure 1.

Some bottom-left companies have traditional dietary supplement brand lines; whereas, others have mail-order business models, which are a major channel in the Japanese market and where customer relationships are a key element. These firms may not have sufficient knowledge to conduct R&D or utilise FFCs regulations.

Companies in the upper left quadrant are primarily food manufacturers with unique and differentiated food products developed through long-term R&D activities using food science and biotechnology. These companies may be able to gain a sustainable competitive advantage through brand equity and promotional capabilities developed over years of corporate activity without having to take advantage of FFCs regulations. In contrast to the upper left quadrant, the upper right quadrant includes firms that entered the market from other industries. They are developing dietary supplement businesses by leveraging their R&D capabilities and conforming to regulations. This suggests that regulations can provide opportunities for new business development. This suggests that regulations can stimulate firms to leverage their capabilities and knowledge to bring innovative products to the market.

The reasons why no firms fall into the lower right quadrant, whose strategy under the FFCs regulation is to curtail their R&D and that rely entirely on the use of external knowledge through SR, are as follows. Public knowledge about the SR of raw materials has commoditized single raw material FFCs with SR. Such products, even if developed at a low cost, would not provide a sustainable competitive advantage. Modular, multifunctional FFCs products that combine multiple ingredients with several SRs, even if they are unique as products, are mere combinations. Therefore, it is difficult to protect intellectual property rights, and they do not provide a sustainable competitive advantage.

The asymmetry between the upper left and lower right quadrants relative to the approximate line indicates that corporate strategies that engage in aggressive R&D do not necessarily involve regulations; whereas, those that leverage regulations involve R&D activities. This finding suggests that knowledge creation through R&D is a prerequisite for innovation through regulation.

Several factors beyond regulation can influence corporate research and development. External environmental factors include market changes and technological advancements. Internal factors unique to each company, such as company size and business composition beyond supplement operations, are also significant. The 15 companies examined in this study include those for which the supplement business is a main operation, as well as companies with annual revenues exceeding one trillion JPY. This indicates a wide range in the share of the nutritional supplement business among the 15 companies studied. Larger corporations generally possess more extensive research and development resources, and they may experience relatively lower compliance costs (particularly in terms of research and development). Conversely, if large corporations possess competitive advantages arising from other business activities (for example, a strong corporate brand or dominant distribution channels), the incentive to gain a competitive advantage through compliance with regulation in the dietary supplement business could be comparatively small. Large corporations might also be hindered in devising and implementing strategies for innovation specifically for their dietary supplement businesses under an overarching corporate strategy. This study acknowledges a limitation in that it did not sufficiently observe and analyse variables other than regulation and product of dietary supplements.

The large companies targeted in this study can create innovative and highly functional FFCs products, increase product diversity, and lead the market by investing in R&D to adapt to regulations. Small- and medium-sized enterprises (SMEs), on the other hand, are less likely to adopt the strategy of adapting to regulations by making burdensome R&D investments. Previous studies have shown that SMEs are new entrants to the FFCs market by utilizing the SR route of FFCs regulation [10]. For SMEs, existing FOSHUs regulations have long been a barrier to entry into new markets. The deregulation from FOSHUs to FFCs has provided market entry opportunities for many companies, including SMEs, and facilitated consumer access to a wide variety of FFCs.

### 6.3. Implications for Regulatory Design

The FFCs system’s regulatory design includes three aspects of Stewart’s framework [26] for regulation that promote innovation. In terms of the first aspect of regulatory stringency, the FFCs system based on a notification system is less burdensome for FOSHUs and can promote innovation. The second aspect, regulatory flexibility, relates to the diversity of regulatory compliance methods, increases incentives for regulatory compliance, and is positive for innovation [14,26]. The FFCs system offers numerous options for health claims in terms of function, ingredients, and wording, thereby providing regulatory compliance incentives for firms. The notification procedure includes two routes, SR and CTs, providing more choices for companies than FOSHUs. This high flexibility provides firms with room for ingenuity, potentially leading to competitive advantages and incentives for regulatory compliance. In terms of the third aspect, information, FFCs regulations aim to reduce the asymmetry in consumer health information [10]. By providing health claims based on scientific evidence, the regulations incentivize firms to provide quality information and also enable them to differentiate their products with unique health benefits.

The interactions between the functional food and pharmaceutical sectors reflect the erosion of industry-specific regulatory boundaries, with companies increasingly incorporating pharmaceutical knowledge into functional foods during the industry convergence process [72]. Some knowledge and technologies, such as quality control, have been transferred from the pharmaceutical industry and are affected by companies’ path dependency [50]. This blending of knowledge from highly regulated domains, facilitated by regulatory design, has implications for innovation and competition in the functional food industry.

The healthcare industry exists at the intersection of the medical and nonmedical sectors, providing fertile ground for innovation. Historically, regulations in the healthcare domain have followed business activities and products rather than preceding them. The regulatory framework is still in a formative and fluid stage, and mutual tensions have existed among stakeholders, such as regulators, producers, consumers, and academia [73,74]. Policymakers have recognized the importance of regulation in this sector, leading to trial-and-error approaches, such as the grey-zone elimination system of the Ministry of Economy, Trade, and Industry in Japan. This case study is an example of the interaction between policy and business and the coevolution of regulation and technology. Rather than imposing highly coercive and inflexible laws, it is more suitable to devise regulations that offer companies strategic choices considering their circumstances. The FFCs system is a pertinent example of an institutional design that fosters innovation.

### 6.4. Prospects for Global Market and Regulation

While this study focused on Japan’s functional food market, it is important to observe and compare environments in other regions and the characteristics of global companies, because the functional food market is expanding globally.

Different countries have varying dietary supplement regulations, and international harmonization has not been achieved [10,31,50]. Japan’s regulations are unique in that they include multitrack health claim regulations (FFCs, FOSHUs, so-called health food) and voluntary GMPs. The FFCs system focuses on a voluntary and flexible system. This is a case with both practical and academic implications.

In contrast, the United States has a comprehensive dietary supplement regulation law (Dietary Supplements Health Education Act (DSHEA)), providing for labelling and GMP compliance. This strong and rigid regulation ensures consumer safety and promotes the growth of the dietary supplement industry.

In emerging countries, the establishment of regulatory frameworks for functional foods is still insufficient. Japan is one of the world’s leading markets for functional foods, and recent regulatory reforms have attracted global attention. This study will also be of interest to policymakers in emerging countries that are building new regulations.

Some companies are expanding their dietary supplement products globally. Global expansion can increase the efficiency of returns on R&D and foster more active R&D investment. On the other hand, inadequate international harmonization of regulation is a factor that increases the cost of regulatory compliance, as it makes compliance with different regulations in different countries inefficient. Global expansion will work both positively and negatively on the relationship between regulation and innovation.

### 6.5. Limitations and Future Perspectives

This study examined the relationship between innovation and regulation in Japan’s functional food industry using the cross-sectional data of 74 functional food products from 15 companies. However, this study had several limitations owing to its specific dataset and context. To gain deeper insights, future research should employ time-course observations and analyses of these cases and compare them with those of other healthcare industries.

The analysis in this study focuses on externally observable indicators, such as sales and sales growth, but other factors, such as profit margins and stock prices, can measure firm performance. Further research is required to consider these additional variables and the various factors that influence firm and product sales. The factors mediating firms’ decisions regarding their degree of regulatory compliance are not well-known. Further investigation into the intricacies of this decision-making mechanism and formulation of a comprehensive model thereof constitutes a prospective avenue for future research. As previously mentioned, this study does not consider factors outside the regulatory framework, such as the impact of market shifts and technological advancements on innovation, or the influence of corporate characteristics outside the dietary supplement segment. This constitutes a limitation of the research.

This study suggests that similar events may occur in other healthcare industries, where companies from different sectors enter the healthcare market, such as health equipment, bedding, fitness, and aesthetic treatments. Regulatory science and convergence research can help design appropriate regulations for these boundary industries, striking a balance between consumer protection, the encouragement of innovation, and cost containment in healthcare. These findings may also apply to the software field, such as medical devices (SaMD), where innovative products are emerging [75].

Future research should explore the relationship between innovation and regulation in various healthcare industries and regions by considering the different regulatory frameworks and their implications for consumer health and healthcare costs.

## 7. Conclusions

This study contributed to the literature by focusing on the Japanese functional food regulatory system and quantitatively analysed the differences in firm behaviour and innovation within and outside the regulatory system. This study discussed firms’ decision-making patterns regarding the compliance with or avoidance of regulations, shedding light on the effectiveness of these regulations in promoting innovation.

This study presents the following primary findings and suggestions. First, a link exists between firms that proactively comply with the FFCs regulations and those that pursue product differentiation and competitive advantage through their R&D. Second, it was suggested that knowledge creation through R&D is a prerequisite for innovation through regulation by analysing a variety of strategies based on company characteristics in terms of accumulated knowledge and the origins of the firms. Finally, it was suggested that regulatory flexibility, offering strategic options for firms even within regulatory constraints, could promote competition and encourage innovation.

In the healthcare domain, regulations play a pivotal role in balancing consumer protection and preventing market failures through innovation activities. In this study, it was suggested that the external environment of regulation may influence the corporate behaviour of the R&D investment strategy and promote the development of differentiated and competitive products. Corporate strategies of regulatory compliance and R&D investment also depend on corporate characteristics. It was, therefore, suggested that flexible regulatory design encourages diverse firms to compete according to their strategies, fostering innovation and creating a healthy healthcare market. This paper offers a valuable case study for discussing the relationship between regulations and innovation, specifically shedding light on the factors by which regulations either foster or inhibit innovation. Furthermore, these insights have practical implications for the design of regulations in the healthcare sector.

## Figures and Tables

**Figure 1 foods-13-03302-f001:**
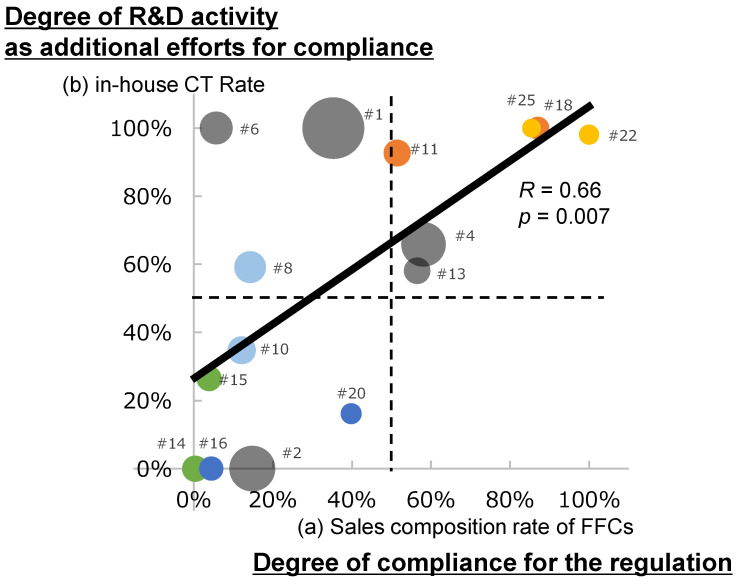
Scattering plots of 15 companies. (a) Sales composition rate of FFCs and (b) in-house CT rate (data shown in Table 2). Circle size indicates sales. The colours of the circle correspond to those of the cell “Company ID” in Table 6. The label numbers represent the company IDs. The bold line shows a linear approximation of (a) and (b). The field is divided into four quadrants based on the respective 50% of (a) and (b), indicated by the dashed lines. (a) The sales composition rate of FFCs was calculated by dividing the total sales of FFCs by the total sales of dietary supplements and suggests the degree of compliance for the FFCs regulation. (b) In-house CT rate was calculated by dividing the total sales of in-house CTs of FFCs by the total sales of FFCs and suggesting the degree of R&D activity as additional efforts for regulatory compliance. The line of linear approximation (*R* = 0.66, *p* = 0.007) indicates that, overall, companies that actively comply with FFCs regulations tend to invest in R&D and implement CTs in-house. However, many companies deviate from the line of approximation, indicating that there are differences in strategies between companies. The difference in attributes between firms in the upper left quadrant (some food companies) and upper right quadrant (some entering companies) suggests that regulatory compliance strategies may depend on firm attributes. The asymmetry between the upper left and lower right quadrants concerning the approximate line suggests that R&D is a necessary condition for innovation that occurs through the utilization of regulations.

**Figure 2 foods-13-03302-f002:**
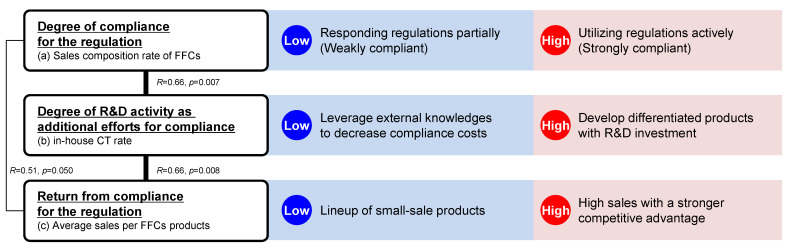
Schematic image of relationship among (a) sales composition rate of FFCs, (b) in-house CT rate, and (c) average sales per FFCs products and company strategies of regulatory compliance and R&D. As shown in the left column, (a) is a metric for compliance with FFCs regulations, (b) is an indicator of R&D activity as additional efforts for compliance, and (c) is a measure of returns on regulatory compliance. Correlation coefficients *R* and *p*-values between each variable are shown between each text box. Positive correlations were observed between these variables, ranging from *R* = 0.51 to *R* = 0.66. As shown in the right column, companies with a more active engagement in the FFCs system generally prefer conducting in-house clinical tests to foster the development of high-sales products. Conversely, as shown in the middle column, companies that only adopt the FFCs system to a limited extent tend to capitalize on external expertise, thereby curating an array of FFCs products that are more economically viable because of reduced compliance costs.

**Table 1 foods-13-03302-t001:** Comparison between dietary supplements (so-called health food and food under the FHCs regulatory system) and pharmaceuticals. Between dietary supplements and pharmaceuticals, there are some similarities in terms of quality, safety, and functionality (or efficacy), such as good manufacturing practices (GMPs) and clinical trials. Compliance costs and the potential for product differentiation, which can act as an incentive for regulatory compliance, vary among the four categories of dietary supplements, depending on the level of regulation. The regulatory complexity of dietary supplements offers companies strategic options to respond to regulations.

Category	The So-Called Health Food	Foods with Health Claims (FHCs)			Pharmaceutical
		Foods with Function Claims (FFCs)		Food for Specified Health Uses (FOSHUs)	
Administrative process	No	Notification		Approval	Approval
Basis of efficacy or functionality	Not allowed to label functions	Systematic review (SR) of functional components (literature of clinical trial)	Product clinical trials	Product clinical trials (large-scale)	Nonclinical studies, clinical trials (3 phases)
Basis of safety	Not required (eating experience assuming safety)	Acceptable by food experience (if the basis of dietary experience is insufficient, it will be based on clinical trials, etc.)		Clinical trials	Nonclinical studies, clinical trials
Quality control regulations	Not mandatory (voluntary GMPs)	GMPs are practically mandatory (described in the guidelines for FFCs)		Not mandatory (voluntary GMPs)	GMPs are mandatory (high level)
Development cost	Extremely low	Low	Middle	High	Extremely high
Product differentiation (incentive to adopt regulation)	Extremely low	Low	Middle	High	Extremely high
	Difficult to differentiate products technically	It is possible to develop similar products with the same ingredients	Uniqueness is ensured by intellectual property rights, etc. Application patents, formulation patents, etc.	In addition to securing intellectual property rights, the cost of administrative processes is a barrier	Secured by ingredient patents, formulation patents, etc.

Note: Foods with Nutrient Function Claims (FNFCs) are omitted from the table, because this system only allows the labelling of their functions in the case of specific nutrients specified in the standard specifications. GMPs: good manufacturing practices.

**Table 2 foods-13-03302-t002:** Properties for FFCs companies (n = 15) and characteristics of their products.

			1	2	3	4	5	6	7	**8**	**9**
ID	Company Name	Main Business of Company	Company Sales	Total Sales of Dietary Supplements	Sales of FFCs	Sales of In-House CTs of FFCs	Number of FFCs	(a) Sales Composition Rate of FFCs	(b) In-House CT Rate	**(c) Average Sales per FFCs Product**	**CAGR of Total Sal** **e** **s of Dietary Supplements**
1	Suntory	Beverages	2970	83.3	29.4	29.4	3	35%	100%	9.8	4.2%
2	DHC	Dietary supplements, cosmetics	ND	45.8	6.8	0.0	10	15%	0%	0.7	1.5%
4	Fancl	Dietary supplements, cosmetics	104	43.1	25.1	16.5	16	58%	66%	1.6	9.1%
6	Meiji	Food, pharmaceuticals	1062	23.6	1.3	1.3	1	6%	100%	1.3	4.2%
8	Asahi	Beverages, foods	2511	22.2	3.2	1.9	12	14%	59%	0.3	16.7%
10	Otsuka Pharmaceutical	Pharmaceuticals, foods	1738	17.3	2.1	0.7	4	12%	35%	0.5	12.9%
11	SetagayaShizen-shokuhin	Dietary supplements	ND	15.7	8.1	7.5	3	51%	93%	2.7	9.9%
13	Ajinomoto	Foods	1359	15.2	8.6	5.0	2	56%	58%	4.3	4.5%
14	Yazuya	Dietary supplements	ND	14.8	0.1	0.0	1	0%	0%	0.1	0.0%
15	Egao	Dietary supplements	ND	13.7	0.5	0.1	2	4%	26%	0.3	−10.4%
16	Kobayashi Pharmaceutical	Pharmaceuticals, daily necessities	166	12.6	0.6	0.0	7	4%	0%	0.1	−1.0%
18	Kyusai	Dietary supplements	ND	11.2	9.8	9.8	2	87%	100%	4.9	−4.0%
20	Taisho Pharmaceutical	Pharmaceuticals	301	9.6	3.8	0.6	5	40%	16%	0.8	9.1%
22	Fujifilm	Chemicals, electronics	2859	8.9	8.9	8.7	5	100%	98%	1.8	34.3%
25	Lion	Toiletries, daily necessities	390	7.6	6.5	6.5	1	85%	100%	6.5	0.3%

Note: The variables are summarized in Section 4. Materials and Methods. ND: no data. Sales units: billion JPY.

**Table 3 foods-13-03302-t003:** Descriptive statistics of the properties of FFCs companies (n = 15) and characteristics of their products.

	**2**	**3**	**4**	**5**	**6**	**7**	**8**	**9**
	**Total Sales of Dietary Supplements**	**Sales of FFCs**	**Sales of In-House CTs of FFCs**	**Number of FFCs**	**(a) Sales Composition Rate of FFCs**	**(b) In-House CT Rate**	**(c) Average Sales per FFCs Product**	**CAGR of Total Sales of Dietary Supplements**
	**(n = 15)**	**(n = 15)**	**(n = 15)**	**(n = 15)**	**(n = 15)**	**(n = 15)**	**(n = 15)**	**(n = 15)**
Min.	7.6	0.1	0.0	1.0	0.3%	0.0%	0.1	−10.4%
Max.	83.3	29.4	29.4	16.0	100.0%	100.0%	9.8	34.3%
Med.	15.2	6.5	1.9	3.0	35.3%	59.2%	1.3	4.2%
Mean	23.0	7.6	5.9	4.9	37.9%	56.7%	2.4	6.1%
S.D.	19.5	8.4	7.8	4.3	32.6%	39.4%	2.7	10.0%

Sales figure units: billion JPY.

**Table 4 foods-13-03302-t004:** Correlation coefficients between indicators for 15 companies using the Foods with Function Claims (FFCs) system.

		2	3	4	5	6	7	8	9
	Variables	Total Sales of Dietary Supplements	Sales of FFCs	Sales of In-House CTs of FFCs	Number of FFCs	(a) Sales Composition Rate of FFCs	(B) In-House CT Rate	(c) Average Sales per FFCs Product	CAGR of Total Sales of Dietary Supplements
2	Total sales of dietary supplements	1							
3	Sales of FFCs	0.77 **	1						
4	Sales of in-house CTs of FFCs	0.72 **	0.95 **	1					
5	Number of FFCs	0.30	0.32	0.10	1				
6	(a) Sales composition rate of FFCs	−0.17	0.41	0.44	−0.06	1			
7	(b) In-house CT rate	0.13	0.47 †	0.60 *	−0.23	0.66 **	1		
8	(c) Average sales per FFCs product	0.48 †	0.67 **	0.79 **	−0.34	0.51†	0.66 **	1	
9	CAGR of total sales of dietary supplements	−0.06	0.13	0.12	0.31	0.38	0.29	−0.12	1

Note: *p*-values of correlation analysis are shown by †, *, and ** (†: *p* < 0.1; *: *p* < 0.05; **: *p* < 0.01). CT = clinical trial.

**Table 5 foods-13-03302-t005:** Multiple regression analysis of in-house CT rate.

Variables	Coefficient *t*	Std. Error	95% Confidence Interval		*t*-Value	*p*-Value
			Lower	Upper		
Constant	0.22	0.12	−0.03	0.47	1.89	0.083
(a) Sales composition rate of FFCs	0.53	0.26	−0.05	1.11	2.00	0.069
(c) Average sales per FFCs product	0.06	0.03	−0.01	0.13	1.98	0.071

*R*-squared: 0.58, adjusted *R*-squared: 0.51; *F*: 8.14; *p* = 0.0058.

**Table 6 foods-13-03302-t006:** The behaviours of 15 companies in response to FFCs regulation. The behaviours are influenced by various factors, including their product lineups, brand strengths, product domains, and technologies. The table is organized to group similar companies into the same row. The colours of the cell “Company ID” correspond to those of the marker in Figure 1.

Company ID	Companies’ Behaviour for FFCs Regulation
#1	Innovation investments develop a small number of large FFCs products. The remaining products are marketed as “so-called health foods” to avoid regulations.
#2	Corporate policy: “offer a variety of products at low prices”; reluctant to comply with FFCs regulations (concentrating on new functional products by utilizing SR).
#4	Corporate policy: “proactively utilise FFCs”; proactively comply with FFCs regulations and differentiate through investment in innovation.
#6	Main focus is on branded sports proteins (noncompliant with regulations). Some FFCs products for sports are sold through in-house CTs.
#8, #10	Has “traditional” supplement brands such as vitamins, minerals, herbs, etc., and is less FFCs-compliant.
#11, #18	Strongly compliant with regulations and investing in innovation to secure competitive advantage with new functional “joint care” supplements.
#13	Leveraged seeds to market amino acids for sports (noncompliant with regulations) and health function amino acids (FFCs-compliant).
#14, #15	Low compliance with FFCs regulations
#16, #20	Low FFCs compliance with many brands of supplements.
#22, #25	Entered the market before the FFCs regulation, strongly compliant with the FFCs regulation, and developed a product with new features related to “weight loss” through its own CT.

## Data Availability

In this paper, we used three types of data sources that are publicly available. (1) Health & Beauty Foods Marketing Summary (Fuji-keizai) is a commercial data book [55]; (2) FFCs data are accessible on the website of the Consumer Affairs Agency [56]; (3) the companies’ data are accessible on the website of each company [57,58,59,60,61,62,63,64,65,66,67,68,69,70,71].

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
