# Peer review of "Driving Innovation Through Regulatory Design and Corporate Behaviour: A Case Study of Functional Food Industry in Japan"

_foods, 2024, doi:10.3390/foods13203302_

Round 1
Reviewer 1 Report
Comments and Suggestions for Authors
The paper “Driving Innovation through Regulatory Design and Corporate Behavior: A Case Study of Functional Food Industry” contributes to the growth of the literature on the functional food regulatory system.
The following items should be revised:
This analysis focus on the Japanese functional food regulatory system, therefore I suggest you write it in the title and purpose.
Methods
“dietary supplement” - I suggest you to describe what supplements - vitamins and minerals, only vitamin or minerals or different, e.g. amino acids?
In the title the authors suggest: "Functional Food" and in methods the authors describe dietary supplement - elaborate why.
Conclusions
The Conclusions are extensive, similar to a summary, and missing two summarising sentences.
Reviewer 2 Report
Comments and Suggestions for Authors
See attached file

Round 2
Reviewer 2 Report
Comments and Suggestions for Authors
Gracias por la reseña. El estudio puede considerarse válido.